# Proline Dehydrogenase (PRODH) Is Expressed in Lung Adenocarcinoma and Modulates Cell Survival and 3D Growth by Inducing Cellular Senescence

**DOI:** 10.3390/ijms25020714

**Published:** 2024-01-05

**Authors:** Sarah Grossi, Elena Berno, Priscilla Chiofalo, Anna Maria Chiaravalli, Raffaella Cinquetti, Antonino Bruno, Maria Teresa Palano, Matteo Gallazzi, Stefano La Rosa, Fausto Sessa, Francesco Acquati, Paola Campomenosi

**Affiliations:** 1Dipartimento di Biotecnologie e Scienze della Vita, DBSV, Università degli Studi dell’Insubria, Via J.H. Dunant 3, 21100 Varese, Italy; sarah.grossi89@gmail.com (S.G.); eberno@uninsubria.it (E.B.); pchiofalo@uninsubria.it (P.C.); raffaella.cinquetti@uninsubria.it (R.C.); antonino.bruno@uninsubria.it (A.B.); francesco.acquati@uninsubria.it (F.A.); 2Unità di Anatomia Patologica, Ospedale di Circolo e Fondazione Macchi, Via O. Rossi 9, 21100 Varese, Italy; annamaria.chiaravalli@asst-settelaghi.it (A.M.C.); stefano.larosa@uninsubria.it (S.L.R.); fausto.sessa@uninsubria.it (F.S.); 3Centro di Ricerca per lo Studio dei Tumori Eredo-Famigliari, Università degli Studi dell’Insubria, 21100 Varese, Italy; 4Laboratorio di Immunità Innata, Unità di Patologia Molecolare, Biochimica, e Immunologia, Istituto di Ricovero e Cura a Carattere Scientifico (IRCCS) MultiMedica, Via Fantoli 16/15, 20138 Milan, Italy; mariateresa.palano@multimedica.it (M.T.P.); 91matteogallazzi@gmail.com (M.G.); 5Centro di Ricerca per l’Invecchiamento di Successo (CRIS), Università degli Studi dell’Insubria, 21100 Varese, Italy; 6Dipartimento di Medicina e Innovazione Tecnologica, DIMIT, Università degli Studi dell’Insubria, Via Guicciardini 9, 21100 Varese, Italy

**Keywords:** proline dehydrogenase, lung cancer, immunohistochemical analysis, adenocarcinoma cell lines, cell-based assays, cellular senescence

## Abstract

The identification of markers for early diagnosis, prognosis, and improvement of therapeutic options represents an unmet clinical need to increase survival in Non-Small Cell Lung Cancer (NSCLC), a neoplasm still characterized by very high incidence and mortality. Here, we investigated whether proline dehydrogenase (PRODH), a mitochondrial flavoenzyme catalyzing the key step in proline degradation, played a role in NSCLC tumorigenesis. PRODH expression was investigated by immunohistochemistry; digital PCR, quantitative PCR, immunoblotting, measurement of reactive oxygen species (ROS), and functional cellular assays were carried out. PRODH expression was found in the majority of lung adenocarcinomas (ADCs). Patients with PRODH-positive tumors had better cancer-free specific and overall survival compared to those with negative tumors. Ectopic modulation of PRODH expression in NCI-H1299 and the other tested lung ADC cell lines decreased cell survival. Moreover, cell proliferation curves showed delayed growth in NCI-H1299, Calu-6 and A549 cell lines when PRODH-expressing clones were compared to control clones. The 3D growth in soft agar was also impaired in the presence of PRODH. PRODH increased reactive oxygen species production and induced cellular senescence in the NCI-H1299 cell line. This study supports a role of PRODH in decreasing survival and growth of lung ADC cells by inducing cellular senescence.

## 1. Introduction

Lung cancer accounts for the leading cause of cancer death worldwide [1]. This is mainly due to the fact that diagnosis occurs at late stages, when tumors are metastasized and surgery is no longer an option, nor is therapy effective [2]. Lung cancer is clinically heterogeneous, comprising Small Cell Lung Cancer (SCLC, accounting for about 15% of lung cancer cases) and Non-Small Cell Lung Cancer (NSCLC, 85% of lung cancer cases), that in turn include two major histotypes, adenocarcinoma (ADC, ≈55% of cases) and squamous cell carcinoma (SCC, ≈45%) [3].

Given the poor survival rate of lung cancer at advanced stages, early diagnosis, prognosis and therapy of NSCLC would greatly benefit from the identification of novel biomarkers.

Since the description of the so-called “Warburg effect” and the most recent findings of the reverse Warburg effect, an altered metabolism has been increasingly recognized to play an important role in tumor promotion and progression [4]. The metabolism of non-essential amino acids (NEAA), such as glutamine or serine, has been shown to be important for cancer cell survival in almost all types of cancer [5].

Proline is an NEAA endowed with essential functions in cell physiology that has been shown to play a role in tumorigenesis [6,7]. Proline can be synthesized in two steps, from glutamate through the intermediate Δ-1-pyrroline-5-carboxylate (P5C), by the activity of P5C synthase (P5CS) and P5C reductase (PYCR, EC 1.5.1.2). This amino acid can also be derived from ornithine by the sequential action of ornithine-δ-aminotransferase (OAT) and P5C reductase [8].

Proline catabolism occurs through specific enzymes. Proline dehydrogenase (PRODH, EC 1.5.5.2, formerly EC 1.5.99.8) catalyzes the first step in the proline degradation pathway [9,10]. This stress-inducible enzyme is localized in the mitochondrial inner-membrane and contains a flavin adenine dinucleotide cofactor (FAD). The regulation of PRODH expression is complex and only partially characterized. It involves several transcription factors with an important role in cancer development, and microRNAs [11,12,13,14].

PRODH can play a dual role in the tumorigenic process, either promoting cell survival through the production of ATP or by inducing reactive oxygen species (ROS)-dependent protective autophagy, as well as by inducing ROS-mediated apoptosis [13,14,15,16,17]. Indeed, PRODH is one of the main proapoptotic effectors of p53, when induced by genotoxic stress [18]. PRODH also plays a role in response to other types of stress, including nutrient-deprivation stress [19,20]. In this context, it must be underlined that its substrate, proline, can be easily retrieved by degradation of the extracellular matrix, in particular collagen, where it accounts, together with hydroxyproline, for 25% of the amino acids [19].

Therefore, PRODH seems capable of influencing the balance between survival and apoptosis, likely depending on the cell type and on the type and severity of stress acting on those cells [11].

Several studies document that PRODH is dysregulated in different types of cancer. In particular, PRODH expression has been studied in kidney, bladder, stomach, colon and rectum and liver, where it was shown to be downregulated compared to normal tissue [21,22]. This, together with evidence that *PRODH* is a known p53 target gene involved in apoptosis [11,21,23] and can suppress tumorigenesis in mouse models, supported the notion of PRODH behaving as an oncosuppressor in these tumors [22]. More recently, however, PRODH expression was shown to sustain invasion and metastatization in breast cancer cells [24] and to promote pancreatic tumor growth [20]. On the other hand, in the MCF7 breast cancer cell line, PRODH downregulation was shown to promote autophagy, whereas its upregulation would promote apoptosis [25,26].

The aim of this study was to investigate whether PRODH could play a role in lung tumorigenesis.

To this aim, PRODH expression in lung cancer was characterized by immunohistochemistry; we tested the possible correlation between the expression of this protein and the expression of known lung cancer markers. Moreover, to elucidate the functions exerted by PRODH in lung cancer, we modulated its expression in ADC cell lines and performed a series of phenotypic assays.

## 2. Results

### 2.1. PRODH Expression Is Elevated in Lung Adenocarcinomas but Not in Squamocellular Carcinomas

To investigate PRODH expression in lung cancer, we performed immunohistochemical analyses on 148 lung cancer samples, of which 135 were NSCLCs and 13 were SCLCs. PRODH was observed as granular immunoreactive deposits in the cytoplasm of tumor cells, with the intensity and percentage of immunoreactive cells variable from case to case. Representative images of PRODH expression are reported in Figure 1A. No immunoreactivity was observed when the primary antibody was substituted with non-immune serum or after pre-absorption of the antibody with 20 nmol of the recombinant protein used to raise it.

Considering a threshold at ≥25% stained cancer cells, PRODH expression was elevated in NSCLC (36.3%) and was never observed in SCLC. In particular, intense and diffuse staining was frequently observed in ADCs (57.14%) and in a small proportion of SCCs (13.85%) (Table 1). The difference in PRODH staining between ADC and SCC cases was statistically significant (*p*-value < 0.0001, Fisher’s exact test). The correlation between the main clinicopathological data and PRODH expression in NSCLC is presented in Table 1. Moreover, the staining was significantly stronger and involved a higher percentage of cells in ADCs, compared to SCCs. Indeed, the mean percentage of immunoreactive cells was 35% (range 0–90%) in ADC and 9% (range 0–75%) in SCC (*p*-value < 0.0001, Student’s *t*-test) (Table 1).

In addition, PRODH expression in lung ADC samples correlated with the stage of the tumors. In particular, PRODH was more expressed in tumors at early stages (pTNM I and II; 66%), compared to tumors at late stages (pTNM III and IV; 23%) (*p*-value = 0.0104, Fisher’s exact test) and it was also more expressed in small tumors (pT1 and pT2; 63,5%) compared to tumors larger than 7 cm or tumors invading other tissues (pT3 and pT4; 0%) (*p*-value = 0.004, Fisher’s exact test). Moreover, the positivity for PRODH was higher in cases without metastasis (pN0; 66%) compared to metastatic cases (pN1 and pN2; 30%) (*p*-value = 0.0420, Fisher’s exact test) (Table 1).

Reverse transcription–quantitative polymerase chain reaction (RT-qPCR) analyses on a subset of samples showed that high levels of protein expression in positive cases were accompanied by an increase in transcript levels. The difference in Delta Ct (Ct_GOI_-Ct_REF_, in which Ct is cycle threshold, GOI is gene of interest, REF is reference gene) median values between PRODH protein-positive and PRODH-negative cases was highly significant (*p*-value = 0.0099, ANOVA test), suggesting concordance between protein and transcript levels (Figure 1B).

Cancer-specific survival rates of patients bearing tumors with high PRODH expression levels were higher when compared to those in patients with low PRODH expression levels, suggesting that PRODH is a favorable prognostic factor in lung ADC, although statistical significance was not achieved due to the relatively small number of cases (Figure 1C). Moreover, our results were confirmed by Kaplan–Meier overall survival curves obtained from the KM-plotter database (https://kmplot.com/, accessed on 11 March 2019), based on a larger cohort (719 cases), where we observed a significant difference in the survival of patients bearing tumors with high levels of PRODH expression, compared to those with low PRODH expression (*p*-value = 0.00048) (Figure 1D).

The association of PRODH expression with Epidermal Growth Factor Receptor (*EGFR*) and *TP53* mutations was also evaluated. In 103 cases, we found that PRODH expression correlated with the presence of *EGFR* activating mutations (*p*-value = 0.0044, Student’s *t*-test) but not with *TP53* mutations (*p*-value = 0.2707 Student’s *t*-test, n = 108).

### 2.2. PRODH Expression Is Low in Lung Adenocarcinoma Cell Lines

We then investigated endogenous PRODH expression in adenocarcinoma cell lines and the consequences of ectopically modulating its expression.

Bioinformatic analysis in the Cancer Cell Line Encyclopedia (https://sites.broadinstitute.org/ccle, accessed on 1 March 2019) allowed us to select lung cancer cell lines with low (A549, NCI-H1299, Calu-6, NCI-H1975, NCI-H2228, NCI-H441, LX-1, SKMES and SK LU-1) or relatively high (HCC827 and its derivative HCC827-GR5, NCI-H1437, NCI-H727 and NCI-H2342) endogenous expression of PRODH.

Actual PRODH expression levels in the cell lines were measured by droplet digital PCR (ddPCR) and Western blot analyses. PRODH was expressed in NCI-H727, HCC827, and its gefitinib-resistant derivatives HCC827-GR5, NCI-H2342 and NCI-H1437 but was almost undetectable in the remaining cell lines (Figure 2), confirming the results from our bioinformatic survey. A direct correlation between transcript and protein levels was observed, confirming our observations in tissues.

### 2.3. PRODH Modulation Affects Survival and Growth of Lung Adenocarcinoma Cells

To compare the ability of different cell lines to survive in the presence or absence of PRODH, we carried out clonogenic assays by transfecting the appropriate constructs and subjecting transfected cells to selection until clones were visible. In all tested cell lines (NCI-H1299, NCI-H1975, Calu-6, NCI-H1437, SK LU-1 and A549), ectopic PRODH expression led to a decrease in cell viability (Figure 3A).

The effects of PRODH ectopic expression on the proliferation of ADC cell lines was evaluated by 3-(4,5-dimethylthiazol-2-yl)-2,5-diphenyltetrazolium bromide (MTT) assay, comparing PRODH-expressing to control clones (empty pcDNA3.1 vector) from NCI-H1299, A549 and Calu-6 cell lines. In all cell lines, PRODH-expressing clones showed a delayed growth that was maximal between days four and six from seeding (Figure 3B). However, we were able to analyze several independent PRODH-expressing clones only in the NCI-H1299 cell line, whereas in all other lung ADC-derived cell lines only very rare PRODH-expressing clones or pools were obtained. This was the case not only for A549 and Calu-6 (Figure 3B), but also for NCI-H1437 and NCI-H1975 cell lines (not shown). Apparently, PRODH expression in vitro was not well tolerated by lung ADC cell lines, except for NCI-H1299 cells.

As PRODH was previously shown to affect breast cancer cells’ growth in 3D but not 2D culture [24], we tested if the growth in 3D was also impaired by carrying out a colony formation assay in soft agar. In NCI-H1299 cells, both the size and the number of colonies formed in soft agar by PRODH-expressing clones were significantly reduced compared to control clones (Figure 3C). In Calu-6 cells, the size of PRODH-expressing clones was reduced compared to control clones, but the number of clones was approximately the same.

### 2.4. PRODH-Expressing Clones from the NCI-H1299 Lung Adenocarcinoma Cell Line Have Increased ROS Levels

In order to investigate the mechanism underlying the reduced cell growth of lung ADC cell lines, we measured ROS production in PRODH-expressing and control clones from the NCI-H1299 cell line. PRODH expression induced a significant increase in ROS levels compared to control clones (Figure 4). ROS increase was prevented by the treatment of PRODH-expressing clones with the ROS scavenger N-acetyl cysteine (NAC, Figure 4). Thus, cells expressing PRODH have higher levels of oxidative stress.

### 2.5. PRODH Expression Does Not Induce Apoptosis in NCI-H1299 Cells

PRODH was initially identified as PIG6, one of the p53 transcriptional targets activated during doxorubicin induced apoptosis [18]. We investigated if the decrease in cell growth observed in PRODH-expressing compared to control clones could be attributed to a greater number of cells undergoing apoptosis. We found a slight, not significant decrease in caspase 3 in PRODH-expressing compared to control clones and the absence of cleaved caspase 3 in all clones from the NCI-H1299 cell line by immunoblot analysis (Figure 5A,B). Similarly, we did not find significant differences either in the number of Trypan blue stained cells (Figure 5C) or in apoptotic cells as analyzed by flow cytometry between the two types of clones (Figure 5D). We concluded that the delayed growth in NCI-H1299 cells ectopically expressing PRODH was not attributable to the induction of apoptosis.

### 2.6. PRODH Expression Induces Cellular Senescence in the NCI-H1299 Lung Adenocarcinoma Cell Line

As PRODH was also shown to induce cellular senescence in different cell types [28,29], we investigated if senescence was the mechanism underlying the decreased cell growth observed in lung ADC cell lines ectopically expressing PRODH.

To this end, we carried out the senescence-associated β-galactosidase (SA-β-gal) assay using clones derived from the NCI-H1299 lung ADC cell line. We observed an increase in SA-β-gal-positive cells in three out of four PRODH-expressing clones. Even taking into account all four PRODH-expressing clones (including the one with no SA-β-gal increase), the average number of SA-β-gal-positive cells in PRODH-expressing clones was significantly higher compared to that of control clones (*p*-value < 0.0001; chi-squared test) (Figure 6A,B). Some of the SA-β-gal-positive cells appeared flattened and vacuolized. The transcript of the cyclin-dependent kinase inhibitor p21 was also significantly increased (Figure 6C).

In keeping with these findings, we observed increased transcript (Figure 6D) and secreted protein (Figure 6E and Appendix A) levels of molecules largely known to be associated with the senescence-associated secretory phenotype (SASP), namely interleukin (IL)-8, monocyte chemoattractant protein-1 (MCP-1) and tumor necrosis factor alpha (TNFα), in three PRODH-expressing clones compared to two control clones of the NCI-H1299 lung ADC cell line.

Finally, we found that the two PRODH-expressing clones had a higher secretion of IL-12, a molecule related to immune cell activation, compared to control clones (Appendix A).

In conclusion, we provide evidence that PRODH affects both the 2D and 3D growth of lung adenocarcinoma cell lines by inducing cellular senescence.

## 3. Discussion

Investigation of the molecular bases of lung cancer, the leading cause of cancer death worldwide [1], is key to finding useful markers for different applications, such as early diagnosis, differential diagnosis and prognosis and to guide therapeutic options. This work focused on the role played by PRODH, a mitochondrial enzyme involved in proline metabolism and playing an important role in the induction of apoptosis and autophagy in lung cancer. PRODH expression has been previously reported to be dysregulated in several types of cancer, including colorectal, renal, mammary, lung and pancreatic carcinomas [20,22,24,25,26,30,31]. In the different tissues, the outcome of PRODH modulation or inhibition has been shown to be quite different. Indeed, whereas in renal cancer and in tumors of the gastrointestinal tract, PRODH appears to induce cell cycle arrest and apoptosis as mechanisms of tumor suppression [21,22], in pancreatic ductal ADC PRODH supports nutrient-deprived pancreatic cell survival and proliferation by utilizing collagen derived proline to fuel the tricarboxylic acids cycle [20]. Moreover, in breast cancer cells PRODH had no effect on 2D in vitro cell growth but supported their 3D growth via ATP production. PRODH also supported metastasis formation in an orthotopic mouse model [24].

Intriguingly, PRODH was also shown to mediate the antitumor effects of antidiabetic drugs such as metformin and troglitazone [13,32].

Thus, PRODH can either suppress or promote cancer development and progression, likely depending on cell and tissue types, and on the type of stress acting on them, by exploiting the use of proline as an intermediate for the conversion to other amino acids and metabolites, whereas the electrons obtained from proline oxidation can be used to produce ATP or ROS to sustain different processes such as cell survival, autophagy, apoptosis or cellular senescence [14,33].

In this work, immunohistochemistry data suggested that PRODH plays a role in lung tumorigenesis, particularly in the ADC subtype, where it appears to improve prognosis in terms of both cancer-specific survival and overall survival. Moreover, in the tumor samples analyzed, we observed that PRODH expression diminished with increasing stage or grading of the ADC samples, suggesting that the function of PRODH in these cancer cells may be related to the maintenance of differentiation and cellular homeostasis. We also found a correlation between PRODH expression and EGFR mutations. This finding is supported by a study in which the transcriptomic analysis of lung cancer samples indicated that PRODH expression levels were increased in the subset of samples carrying EGFR mutations [34].

A comparison of the PRODH transcript levels described in silico with experimental data showed a good correlation. We also found a perfect correlation between PRODH transcript and protein levels, both in tumor tissues analyzed by immunohistochemistry and in lung ADC cell lines, suggesting that PRODH is mainly regulated at the transcriptional level.

Several lung ADC cell lines with low endogenous PRODH expression levels were used to test the clonogenic ability after transfection with an expression construct encoding wild-type PRODH or empty vector as control. In all tested cell lines, transfection with the PRODH expression construct led to a decrease in clonogenic ability compared to the same cells transfected with an empty vector.

Although we were able to obtain PRODH-transfected clones from all cell lines tested (Figure 3A), when single clones were expanded in culture, only in the NCI-H1299 cell line did we obtain several PRODH-expressing clones, whose expression was very stable over time and upon freezing and thawing. By contrast, attempts to obtain clones stably expressing PRODH from the other cell lines were overall unsuccessful, since most clones that were resistant to the selective agent (G418) lost the expression of the transgene with time and only few clones could be analyzed in the described assays for the A549 and the Calu-6 cell lines. This may be due to the features of the expression vector where the PRODH coding sequence was cloned (pcDNA3.1), as transgene expression is under the control of the strong cytomegalovirus early promoter, which has often been shown to lead to unstable transgene expression [35]. In this case, a possible solution may be to clone the PRODH coding sequence into a vector with a different promoter or a vector allowing inducible transgene expression.

In any case, although the conclusions drawn in this work are based on few cellular models, most cancer cell lines used apparently showed extreme sensitivity to PRODH overexpression in vitro, thus reinforcing the notion of an oncosuppressive role for this gene in lung cancer.

The observation that a high percentage of NCI-H1299 stably transfected clones retained PRODH expression might suggest that the genetic background of the transfected cell lines could also affect their “tolerance” to high PRODH expression levels. In this regard, the NCI-H1299 cell line is *TP53* null, whereas the A549 cell line is *TP53* wild-type and the Calu-6 cell line carries a heterozygous nonsense mutation, producing a protein lacking the tetramerization domain; it is expected that the remaining wild-type allele maintains p53 function [36]. It may be speculated that PRODH expression and the consequent oxidative milieu could be stably maintained in vitro only in the absence of functional p53, which would otherwise be activated by this type of stress to induce apoptosis.

In our experiments, PRODH-expressing clones from the NCI-H1299, A549 and Calu-6 cell lines showed delayed growth compared to control clones in both 2D and 3D cultures. Moreover, PRODH-expressing clones had increased ROS levels, and a higher percentage of cells showed several features of cellular senescence, including the expression of SA-β-gal, an increase in the levels of the cell cycle inhibitor p21 and some cytokines of the SASP [37].

PRODH was previously identified as one of the genes differentially expressed in senescent and apoptotic cells after treatment with different doses of etoposide [28]. Although senescence and the induced genes were regulated by p53, it was shown that PRODH expression alone was able to induce cellular senescence, even in p53 null cells [29]. Thus, our data confirm these observations in lung cancer cell lines as well, providing support to the general role of PRODH in the induction of cellular senescence.

Cellular senescence is recognized to play different roles in cancer: on the one hand, it can exert a tumor-suppressive role by inducing a persistent growth arrest; on the other hand, several features of senescent cells, including cell survival and the secretion of cytokines of the SASP, can promote tumor growth, progression, resistance to therapy and relapse [38,39]. Whether the induction of full-blown senescence can be obtained by sole PRODH expression or whether PRODH-induced senescence may represent a tumor-suppressive mechanism needs further evaluation. Data from immunohistochemistry experiments and Kaplan–Meier curves suggest that at least for patients with low-stage, naïve lung ADCs, PRODH has indeed a protective role. This does not exclude the possibility that during patient treatment, PRODH expression may instead become prognostically unfavorable. Further studies are needed to investigate this aspect.

Intriguingly, patients’ age could also influence PRODH-induced processes. In epigenome-wide methylation–age interaction analysis aimed to identify age-specific, prognosis-associated epigenetic biomarkers in NSCLC patients, *PRODH* promoter methylation was the only identified factor that interacted with age to affect prognosis. Indeed, low methylation of *PRODH* CpG island (where high levels of PRODH expression are expected) was proposed to benefit the survival of elderly lung ADC patients (age > 65 years), whereas it represented a negative prognostic factor for younger patients [40]. It is worth mentioning that senescence outcome also depends on age [39].

In another study investigating the effects of PRODH expression in lung ADC cell lines, Liu et al. found that PRODH promoted cell growth compared to control cells [31]. It must be underlined that a different construct and a different promoter driving the expression of the transgene were used in this work compared to the present study. Thus, the observed discrepancy may be attributable to different levels of expression attained in the two systems. The use of an inducible construct may help to understand if this is indeed the case. Given the ability of PRODH to influence various processes such as survival, apoptosis and senescence via ROS or ATP production, the observation of different outcomes with different expression levels is not unexpected.

Also in the work by Liu et al., PRODH ectopic expression led to an increase in some cytokines that are also related to senescence, namely Cxcl1, Lcn2 and IL17c, through the activation of the NFkB inflammatory pathway [31]. The increase in several chemokines and cytokines that we and Liu et al. observed suggests that PRODH expression may regulate the infiltration of both innate and adaptive immune cells in lung tumors. This hypothesis is currently under investigation, as it may represent a further mechanism by which PRODH can exert either pro- or anti-tumorigenic functions.

Our study has some limitations. It is an in vitro study, and experiments were carried out using a limited number of lung ADC cell lines. However, the results obtained warrant further in vivo experimentation with mouse models.

In conclusion, the results described in this study suggest that PRODH plays a role in lung cancer tumorigenesis by inducing cellular senescence. The significance and possible exploitation of cellular senescence for lung cancer therapy will be further investigated.

## 4. Materials and Methods

### 4.1. Samples for Immunohistochemical Analysis

The study was performed on formalin-fixed paraffin-embedded (FFPE) samples of lung cancer collected by the Pathology Department of the “Ospedale di Circolo” in Varese from 1996 to 2015.

Hematoxylin–eosin tumor sections were revised and all tumors were classified according to the criteria of the World Health Organization (WHO) classification system, 4th edition [41]. The tumor stage was assessed using the tumor node metastases system (TNM), as defined by the American Joint Committee on Cancer [42].

A total of 135 NSCLC and 13 SCLC cases were included in this study. Among NSCLCs, 70 were ADCs and 65 were SCCs. Healthy lung tissue, in which some pneumocytes showed intense immunoreactivity, was used as a control.

The study was approved by the institutional review board ethics committee. Subjects were recruited within a clinical protocol by the “University of Insubria” and “Ospedale di Circolo Fondazione Macchi” and approved by the institutional Ethical Committee (approval no. 0037527 of 30 September 2014). The subjects recruited were required to sign their informed consent. The study was carried out in accordance with the Declaration of Helsinki (1975), and as revised in 2013.

### 4.2. Immunohistochemical Analyses

Immunohistochemical analyses were carried out on 3 μm thick formalin-fixed, paraffin-embedded sections, deparaffinized and rehydrated through Bioclear solution, decreasing alcohol series, and then water. After washing in Tris-buffered saline (TBS) pH 7.4, endogenous peroxidase activity was blocked with 3% aqueous hydrogen peroxide for 15 min, followed by washes in TBS + 0.2% Triton (*v*/*v*). Antigen retrieval was performed by incubating slides in a solution of trypsin at a final concentration of 0.5 mg/mL (starting from a 50 mg/mL stock) in TBS for 20 min at 37 °C. For the detection of the immunohistochemical expression of PRODH protein, we initially tested a commercial anti-PRODH antibody raised in rabbit (Prestige anti-PRODH antibody, code HPA020361, Sigma-Aldrich, Milan, Italy; stock 0.05 mg/mL, working dilution 1:100) in parallel with a custom antibody, raised in rabbit using PRODH recombinant protein encompassing amino acids 176–572 as immunogen [43] (Davids Biotechnologie, Dabio, Regensburg, Germany, kind gift from Prof. Pollegioni, University of Insubria, Varese, Italy; stock 1.2 mg/mL). Subsequently, all the immunohistochemical analyses were performed with the custom rabbit polyclonal antibody (Davids Biotechnologie, Dabio, Regensburg, Germany), at a 1:100 dilution in 1% normal goat serum in TBS, with overnight incubation at 4 °C. Negative controls were performed by substituting the primary antibody with non-immune serum or by the pre-absorption of antibody with 20 nmol of the recombinant protein used for raising the antibody (kind gift from Prof. Pollegioni, University of Insubria, Varese, Italy). Sections were then washed in TBS + 0.2% Triton (*v*/*v*) and the signal was detected with the UltraVision Quanto detection system HRP DAB (ThermoFisher Scientific, Milan, Italy) according to the manufacturer’s protocol. Nuclei were counterstained with Harris hematoxylin and after rinsing in running water, sections were dehydrated and embedded in Pertex (Kaltek Srl, Padua, Italy).

A case was considered positive for PRODH staining when at least 25% of tumor cells showed cytoplasmic immunoreactivity.

### 4.3. Cell Culture and Constructs

The cell lines used in this study, their culture conditions and their sources are described in Table 2. All media and reagents for cell culture were from Euroclone (Pero, Milan, Italy), unless otherwise specified.

Cells were incubated at 37 °C in 95% humidity and 5% CO_2_ atmosphere. A new batch of frozen cells was thawed on average 2 weeks before each experiment to provide the necessary number of cells on the day of the experiment. Upon thawing, the cell lines were used within 10 passages, and then cells were discarded and a new batch was thawed. The cell lines used for the experiments were authenticated at Eurofins (Milan, Italy).

To evaluate the presence of mycoplasma, cells were routinely screened by using the nested PCR method described by Tang et al. [44].

The PRODH overexpressing construct, carrying the wild-type PRODH coding sequence under the early cytomegalovirus promoter, was generated by cloning the full-length cDNA of PRODH (corresponding to transcript isoform 1, NM_016335) into the pcDNA3.1 vector. For stable transfections, cells were seeded in a 24-well or 6-well multiwell plate, to reach 70% confluency on the following day, when transfection was carried out using 0.4 µg (24-well) or 1.2 µg (6-well) of DNA and Fugene HD transfection Reagent (Promega, Milan, Italy), according to the manufacturer’s instructions. Then, 24 h later, cells were split into one or more Petri dishes and after a further 24 h the appropriate concentration of G418 (Aurogene, Rome, Italy) was added for the selection of cell clones. In particular, for G418 selection, 300 μg/mL (NCI-H1299), 320 μg/mL (A549), 900 μg/mL (Calu-6), 750 μg/mL (SK LU-1), 400 μg/mL (NCI-H1437) and 600 μg/mL (NCI-H1975) were used, as determined by preliminary cytotoxicity curves. Selective growth medium was substituted every 3–4 days until clones were large enough to be isolated and expanded. Single or pooled clones were frozen in liquid nitrogen and the expression of the transgene was checked by immunoblot analysis.

### 4.4. RNA Extraction from FFPE Tumors and qPCR Analyses

Total RNA from FFPE tissues was extracted using the RecoverAll Total Nucleic Acid Isolation Kit (Ambion by Life Technologies, Milan, Italy) following the manufacturer’s instructions. RNA samples were quantified with a Qubit RNA assay kit (Life Technologies, Milan, Italy) on a Qubit instrument (Life Technologies, Milan, Italy) and run on agarose gel for quality control.

cDNA was prepared from 500 ng of RNA using the iScript select cDNA synthesis kit (Biorad, Milan, Italy), and a reverse primer specific for PRODH (5′-TGGTATTGCTTGTCCCGCTT-3′) and B2M (encoding beta-2-Microglobulin, 5′-GTCCCGGCCAGCCAGGTCC-3′).

For real-time quantitative PCR (qPCR), primer pairs specific for PRODH (PRODH-F 5′-GCAGAGCACAAGGAGATGGA-3′, PRODH-R 5′-TGGTATTGCTTGTCCCGCTT-3′) and B2M (B2M-F 5′-AGGCTATCCAGCGTACTCCA-3′ and B2M-R 5′-ATGGATGAAACCCAGACACA-3′) were used.

Gene expression analysis was performed in triplicate using a CFX96 thermal cycler (Biorad, Milan, Italy) and the iTAQ Universal Sybr Green Supermix (Biorad, Milan, Italy). Blank controls, in which distilled water was used instead of cDNA, were included in each analysis. Melting curve analysis was performed to ensure that single amplicons were obtained for each target.

The difference (Delta Ct) between the Ct obtained for PRODH (GOI) and that for B2M (REF) was calculated for each sample. The smaller the value of the Delta Ct, the higher the expression levels of PRODH.

### 4.5. RNA Extraction from Cell Lines and Clones, Digital PCR and Quantitative PCR Analyses

Total RNA was extracted from cell lines with TriReagent (Sigma-Aldrich, Milan, Italy), according to manufacturer’s instructions. RNA samples were quantified with a NanoDrop 2000c (ThermoFisher by Life Technologies, Milan, Italy) and run on an agarose gel for quality control.

cDNA was obtained from 500 ng of RNA by using the iScript cDNA synthesis kit (Biorad, Milan, Italy).

For ddPCR, 15 ng cDNA was used in a 20 μL reaction, adding 10 μL of QX200 EvaGreen ddPCR Supermix (Biorad, Milan, Italy), primer pairs specific for PRODH and B2M (described in Section 4.4) and nuclease-free water to volume. Two blanks, in which distilled water was used instead of cDNA, were included in each analysis.

Each 20 µL reaction was loaded into a well of a droplet generation cartridge (Biorad, Milan, Italy) and 70 µL of QX200 Droplet generation oil (Biorad, Milan, Italy) was added into the appropriate wells. The cartridge was loaded into the QX200 Droplet Generator (Biorad, Milan, Italy) to generate the droplets, which were transferred to a 96-well plate with a Rainin multichannel pipette. The plate was sealed with Pierceable foil (Biorad Milan, Italy) and put in a T100 thermal cycler (Biorad, Milan, Italy).

Cycling conditions were 95 °C for 5 min, followed by 40 cycles of 95 °C for 30 s and 60 °C for 1 min, then signal stabilization steps (4 °C for 5 min, 90 °C for 5 min) and final hold at 4 °C. The ramp rate was 2 °C/s. After PCR, plates were loaded into the QX200™ Droplet Reader (Biorad, Milan, Italy) for detection.

For qPCR, analyses were performed in triplicate using a CFX96 thermal cycler (Biorad, Milan, Italy) and the iTAQ Universal Sybr Green Supermix (Biorad, Milan, Italy). Blanks, containing distilled water instead of cDNA, were included. Relative mRNA quantification was obtained by applying the 2^-DeltaDeltaCt method [45], after checking the efficiencies of the assays, using B2M for normalization purposes. Melting curve analysis was performed to ensure that single amplicons were obtained for each target. Primer sequences (5′-3′) were:

CDKN1A (p21)-F CTGGAGACTCTCAGGGTCGAAA;

CDKN1A (p21)-R GATTAGGGCTTCCTCTTGGAGAA;

CCL2 (MCP1)-F AGAGGCTGAGACTAACCCAGA;

CCL2 (MCP1)-R TTTCATGCTGGAGGCGAGAG;

TNFA (TNFα)-F TGCACTTTGGAGTGATCGG;

TNFA (TNFα)-R TCAGCTTGAGGGTTTGCTAC;

CXCL8 (IL-8)-F CCTGATTTCTGCCAGCTCTGTG;

CXCL8 (IL-8)-R GTGGTCCACTCTCAATCACTCTC.

Primers for PRODH and B2M were the same as described in Section 4.4.

### 4.6. Immunoblotting

Protein extracts were obtained by mechanically scraping the cells from 100 mm plates in phosphate-buffered saline (PBS) supplemented with 5 mM ethylenediaminetetraacetic acid (EDTA, Euroclone, Milan, Italy). Cells were counted and resuspended in radioimmunoprecipitation assay buffer (RIPA Buffer: 150 mM NaCl, 50 mM Tris-HCl pH 7.5, 1% Igepal CA630, 0.1% SDS, 0.5% Sodium deoxycholate) supplemented with protease inhibitors [phenylmethylsulfonyl fluoride (PMSF), benzamidine, aprotinin and leupeptin (all from Sigma Aldrich, Milan, Italy)]. Samples were incubated on a rotating wheel at 4 °C for 50 min, and then the insoluble fraction was removed by centrifugation. Protein concentration was evaluated using the Quick Start Bradford 1× Dye Reagent (Biorad, Milan, Italy) following the manufacturer’s instructions, using bovine serum albumin to build a standard curve.

Alternatively, cell lysates were obtained after cell detachment from cell culture vessels. Cells were counted and resuspended in Laemmli Sample Buffer 2×, using 1µL of buffer every 2 × 10^4^ cells. The samples were lysed by incubation at 95 °C for 3 min followed by vortexing, repeating these steps three times.

For sodium dodecyl sulphate-polyacrylamide gel electrophoresis (SDS-PAGE), 50 µg of extract or 12 µL of cell lysate were used. Proteins were transferred onto nitrocellulose membranes (Amersham Hybond ECL, GE Healthcare Life Sciences by Euroclone, Milan, Italy) using the Mini-PROTEAN Tetra System (Biorad, Milan, Italy). After blocking with 4% non-fat milk in PBS-T (0.1% Tween20 in PBS) and incubating with the appropriate primary and secondary antibodies, signals were detected with the Amersham ECL (enhanced chemiluminescence) Prime Western Blotting Detection Reagent (GE Healthcare, Milan, Italy), using an Odyssey LI-COR FC imaging system (Carlo Erba Reagents, Milan, Italy).

Primary antibodies were rabbit polyclonal anti-PRODH (stock 0.35 mg/mL, working dilution 1:1000, #SAB1303113, Sigma-Aldrich, Milan, Italy) and anti-caspase 3 (stock 49 μg/mL, working dilution 1:1000, #9662, Cell Signaling Technologies by Euroclone, Milan, Italy); anti-glyceraldehyde-3-phosphate dehydrogenase (GAPDH; stock 0.66 mg/mL, working dilution 1:3000, #E-AB-40337, Elabscience, Milan, Italy) and mouse monoclonal anti-alpha-tubulin (stock 1 mg/mL, working dilution 1:1500, #MAB-94264, Immunological Sciences, Rome, Italy) were used for normalization.

Secondary antibodies were stabilized goat anti-rabbit monoclonal antibody conjugated with horseradish peroxidase (HRP, stock 10 μg/mL, working dilution 1:900, ThermoFisher Scientific, Milan, Italy) and stabilized goat anti-mouse HRP-conjugated monoclonal antibody (stock 10 μg/mL, working dilution 1:900, ThermoFisher Scientific, Milan, Italy). All antibodies were diluted in 2% non-fat milk in PBS-T.

### 4.7. Colony Formation Assay

Transfection was performed as described above, except equimolar amounts of PRODH construct and empty vector were used, and salmon sperm DNA was added to obtain the same total quantity in micrograms and use the same volume of transfection reagent. The 60 mm diameter cell culture Petri dishes were used for these experiments. After 15 days of selection, clones were washed with PBS, fixed with 2% glutaraldehyde in PBS, and stained with 1% methylene blue (Sigma Aldrich, Milan, Italy) in 50% ethanol solution for 30 min directly in the culture dishes. After two additional washes with PBS, dishes were rinsed with deionized water and dried. Clone density was then evaluated in three biological replicates for most cell lines.

### 4.8. Cell Proliferation Assays

Control or PRODH-expressing clones from the NCI-H1299, Calu-6 and A549 ADC cell lines were plated onto 96-well plates at 1 × 10^3^, 1.5 × 10^3^ and 0.7 × 10^3^ cells/well, respectively. An MTT assay was used to monitor cell proliferation for a time frame of 7 days. At each time point, the medium was discarded, and cells were washed with PBS. Then, 100 μL of MTT (0.5 mg/mL in serum-free medium; Across Organics by Carlo Erba, Milan, Italy) was added and incubated for 2.5 h at 37 °C. After a wash with PBS, 80 µL of dimethyl sulfoxide (DMSO; Euroclone, Milan, Italy) was added to each well. Following 30 min of incubation with gentle shaking at room temperature to dissolve MTT crystals, the absorbance at 590 nm was read with an Infinite 200 plate reader (Tecan, Cernusco sul Naviglio, Italy).

### 4.9. Soft Agar Assay

Cells from single control or PRODH-expressing clones from NCI-H1299 or pools from Calu-6 cells were suspended in 0.3% agar (stock solution 2% agar in H_2_O, sterilized by autoclaving) in Roswell Park Memorial Institute (RPMI) 1640 cell culture medium containing 20% fetal bovine serum (FBS) at a density of 1 × 10^2^ cells/well, and plated on solidified agar (0.6% agar in RPMI 1640 culture medium containing 20% FBS) in 12-well dishes. Three replicates were prepared for each condition and each cell line.

Plates were incubated for 16 days at 37 °C in 5% CO_2_. The growth of colonies was monitored using a light microscope (TIEsseLab, Milan, Italy), and their number and size were evaluated. The reported results are the average of three independent experiments.

### 4.10. ROS Measurement

Four PRODH-expressing and two control NCI-H1299 clones were seeded in triplicate in a dark, clear-bottomed 96-well microplate (Greiner, Euroclone, Milan, Italy) to reach 80% confluence four days after seeding. On the fourth day, cells were subjected to serum starvation for 10 min and then incubated for 30 min at 37 °C, 5% CO_2_ in the dark with a 10 µM 2′,7′-dichlorofluorescein diacetate (DCFDA) solution (Sigma Aldrich, Milan, Italy) prepared in RPMI without serum and without phenol red. After incubation, the medium with DCFDA was removed and replaced with medium without phenol red and without serum. ROS production was measured by reading the fluorescence (excitation 485 nm, emission 530 nm) in a microplate reader (TECAN Infinite 200 PRO, Tecan, Cernusco sul Naviglio, Italy). To obtain the negative control, PRODH-expressing clones were treated with 15 mM NAC for 45 min before and during incubation with DCFDA solution. Positive control was obtained by incubating cells from control clones with a DCFDA solution added with 100 µM H_2_O_2_. The results were normalized by the total number of cells in each well. Cell number was evaluated by constructing a standard curve with MTT using different numbers of cells. The results shown are the average of three independent experiments.

### 4.11. Cell Viability Evaluation by Trypan Blue Exclusion Assay

To compare the number of dead cells in four PRODH-expressing clones and two control clones, we performed an initial analysis by staining cells with Trypan Blue dye (Sigma Aldrich, Milan, Italy). Cells were seeded in a 6-well plate to reach 80% confluence after four days from seeding. On day four, cells were detached and an aliquot of the cellular suspension was diluted 1:2 with Trypan blue dye; the cells were counted with Burker chamber using a light microscope (TIEsseLab, Milan, Italy). The test was repeated twice, and at least 700 cells were counted for each clone.

### 4.12. Detection of Apoptosis by Flow Cytometry

PRODH-expressing and control NCI-H1299 clones were seeded in a T25 flask to reach 80% confluence 48 h after seeding. After 48 h, cells were detached with EDTA 1 mM and centrifuged. Cells were resuspended with Annexin V Binding Buffer (BD Biosciences, Milan, Italy), following the manufacturer’s instructions. Cells were then stained with 1 μg/mL Propidium Iodide (Sigma Aldrich, Milan, Italy) and Annexin-V-APC (1:100, Immunotools, Friesoythe, Germany), and apoptosis was evaluated by cytofluorimetric analysis using a BD Biosciences LSR Fortessa X-20 flow cytometer (BD Biosciences, Milan, Italy), followed by data analysis with the FlowJo software, version V10 (BD Biosciences, Milan, Italy).

### 4.13. Senescence-Associated β-Galactosidase Assay

Cellular senescence was evaluated by quantifying cells positive for SA-β-gal expression at pH 6.0 as described by Debacq-Chainiaux et al. [46]. PRODH-expressing and control clones from the NCI-H1299 cell line were seeded to have the same confluence at five days post-seeding, when the difference in cell growth was maximal, as observed in MTT assays. Five days after seeding, cells were washed with PBS two times and then fixed with a fixing solution (2% formaldehyde (*v*/*v*) and 0.2% glutaraldehyde (*v*/*v*) in PBS buffer) for 5 min. Cells were then washed twice with PBS and incubated for 12–16 h at 37 °C without CO_2_ with a staining solution (40 mM citric acid/Na phosphate buffer, 5 mM K_4_[Fe(CN)_6_] 3H_2_O, 5 mM K_3_[Fe(CN)_6_], 150 mM sodium chloride, 2 mM magnesium chloride, 1 mg/mL X-gal in distilled water) containing the chromogenic substrate X-gal (5-bromo-4-chloro-3-indolyl-β-D-galactopyranoside). Cells were then washed with PBS followed by incubation with methanol and plates were left to dry. SA-β-gal activity was determined by observing the cells under the light microscope (TIEsseLab, Milan, Italy). Three independent experiments were carried out. For each clone and each replicate, images from at least 10 randomly selected microscopic fields were taken. A minimum of 500 total cells were analyzed and the percentage of SA-β-gal positive cells was determined.

### 4.14. Generation of Conditioned Media

Conditioned media from two PRODH-expressing and two control clones from the NCI-H1299 cell line were obtained as follows. Briefly, cells were seeded at a density of 1 × 10^6^ cells into 100 mm plates to reach 80% confluence. Then, cells were washed with serum-free medium for 30 min and then incubated in serum-free medium for 72 h. The conditioned media were subjected to two steps of centrifugation to eliminate cell debris, then samples were filtered and concentrated using 3 kDa Amicon 4-Ultracentrifugal filter units (Merck Millipore, Milan, Italy). Total protein content in conditioned media was determined using the Quick Start Bradford 1× Dye Reagent (Biorad, Milan, Italy) following the manufacturer’s instructions, using bovine serum albumin to build a standard curve and reading the absorbance with an Infinite 200 plate reader (Tecan, Cernusco sul Naviglio, Italy).

### 4.15. Analysis of the Senescence-Associated Secretory Phenotype by Protein-Membrane Arrays

Conditioned media from two PRODH-expressing and two control clones from the NCI-H1299 cell line were analyzed with the C-series Human Cytokine Antibody Array C100 (Human Cytokine Array C6 and Human Cytokine Array C7, RayBiotech, Inc., Norcross, GA, USA).

Secretome profiling was performed using 50 µg of total protein from the conditioned media after processing as described above. After incubation, the chemiluminescent signals developed by HRP were detected using an Alliance Q9 (Uvitec) instrument. Signals were quantified by ImageJ software, version 1.53q, National Institute of Health. Two biological replicates were performed for each clone.

### 4.16. Statistical Analysis

The difference in PRODH staining in the samples analyzed by immunohistochemistry was evaluated by Fisher’s exact test; the difference in PRODH expression among ADC and SCC samples was determined using Student’s *t*-test. The correlation between PRODH expression levels and the presence of EGFR activating mutations or p53 mutations was evaluated using Student’s *t*-test. The failure time according to PRODH in Kaplan–Meier curves was evaluated by a chi-squared test. The difference in the distribution of Delta Ct (GOI-REF) values between immunohistochemically PRODH-positive and -negative cases was evaluated by an ANOVA test. The statistical significance of differences in cellular senescence, as determined by the SA-β-gal, was calculated with the chi-squared test. The difference in cytokine secretion obtained from this was determined by Student’s *t*-test. A *p*-value less than 0.05 will be considered statistically significant.

All statistical analyses were carried out with GraphPad Prism software, version 9.

## Figures and Tables

**Figure 1 ijms-25-00714-f001:**
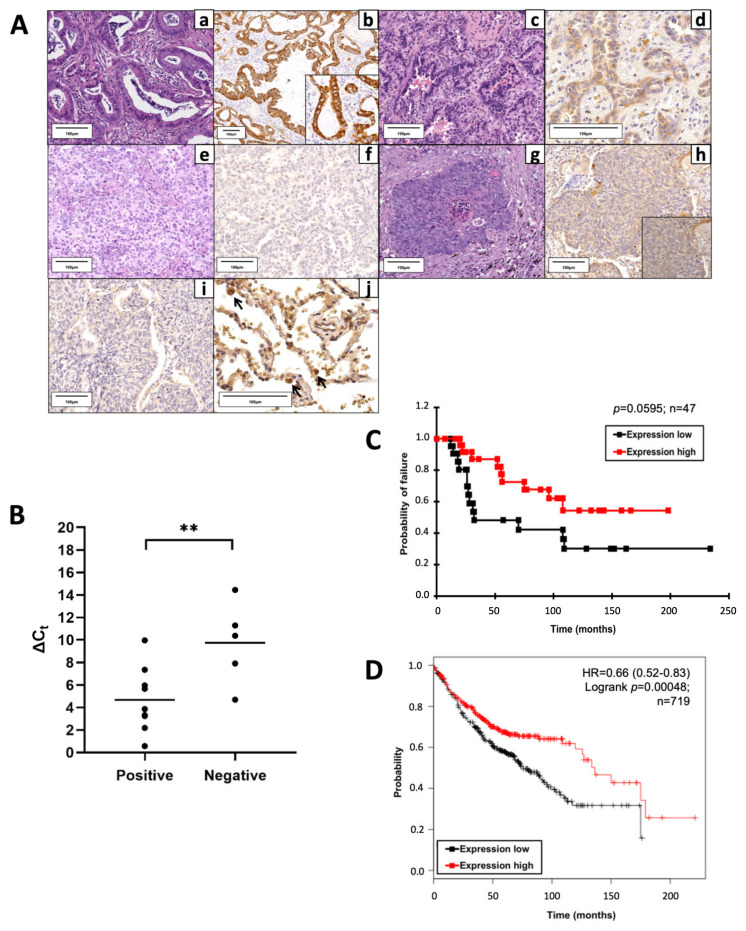
Proline dehydrogenase (PRODH) is expressed in the majority of lung adenocarcinomas (ADC) and represents a prognostically favorable factor. (**A**) Representative results of PRODH immunostaining. Lung ADC: (**a**) Acinar ADC (hematoxylin–eosin, 200×) with (**b**) abundant and intense cytoplasmic granular PRODH immunoreactivity (diaminebenzidine (DAB)-hematoxylin, 100×, particular 400×); (**c**) Acinar ADC (hematoxylin–eosin, 200×) with (**d**) weak cytoplasmic PRODH immunoreactivity (DAB-hematoxylin, 400×); (**e**) predominantly solid lung ADC (hematoxylin–eosin, 200×) with (**f**) no PRODH immunoreactivity (DAB-hematoxylin, 200×). Lung squamocellular carcinoma (SCC): (**g**) SCC (hematoxylin–eosin, 200×) displaying (**h**) weak, diffuse cytoplasmic PRODH staining (DAB-hematoxylin, 200×, inset 400×); (**i**) SCC devoid of PRODH immunoreactivity (DAB-hematoxylin, 200×). (**j**) Healthy lung parenchyma, showing rare PRODH-positive cells, corresponding to type II pneumocytes and Clara cells (black arrows, DAB-hematoxylin, 400×). The bars indicate 100 μm. (**B**) The increase in PRODH protein levels in immunohistochemistry is paralleled by an increase in transcript levels in quantitative PCR (qPCR). Positive: lung ADC cases with elevated expression of PRODH protein; negative: lung ADC cases with low or no expression of PRODH protein. A small Delta Ct (cycle threshold) value indicates high transcript levels (Ct for PRODH similar to that of the reference gene). Horizontal lines indicate the mean value. Asterisks indicate that there is a significant difference (** *p*-value = 0.0099, ANOVA test). (**C**) Kaplan–Meier curves reporting cancer-specific survival for the cohort under study. Cancer-specific survival for ADC samples with high or low PRODH expression levels from this study (cutoff value was 25%, *p*-value = 0.0595, chi-squared test). (**D**) Kaplan–Meier curves reporting overall survival from the KMplotter database for lung ADC samples. Overall survival for ADC samples with high or low PRODH expression (*p*-value = 0.00048, chi-squared test) [27].

**Figure 2 ijms-25-00714-f002:**
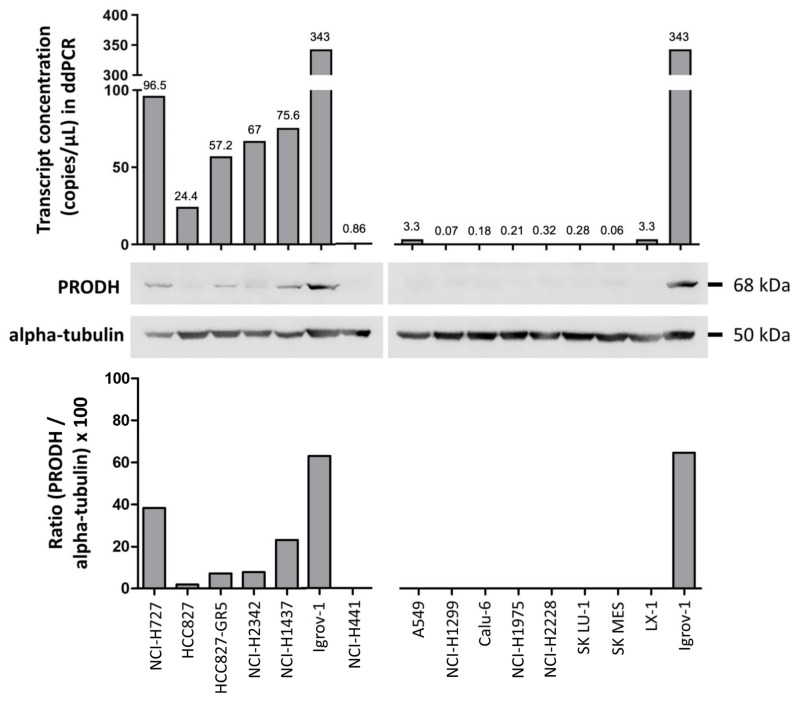
Evaluation of endogenous PRODH expression in a panel of lung ADC cell lines. Expression analysis was carried out at the transcript level using droplet digital PCR. Results are expressed as copies of transcript/microliter of PCR reaction. As the same RNA (ribonucleic acid) and cDNA (complementary deoxyribonucleic acid) volumes were used for all cell lines, these data are directly comparable. Western blots of extracts from the indicated cell lines were detected with PRODH antibody and with alpha-tubulin for normalization. The graph below the Western blots represents the relative expression of PRODH protein in percentage after normalization (ratio PRODH/alpha-tubulin × 100). The uncropped original western blots have been submitted in the Appendix A.

**Figure 3 ijms-25-00714-f003:**
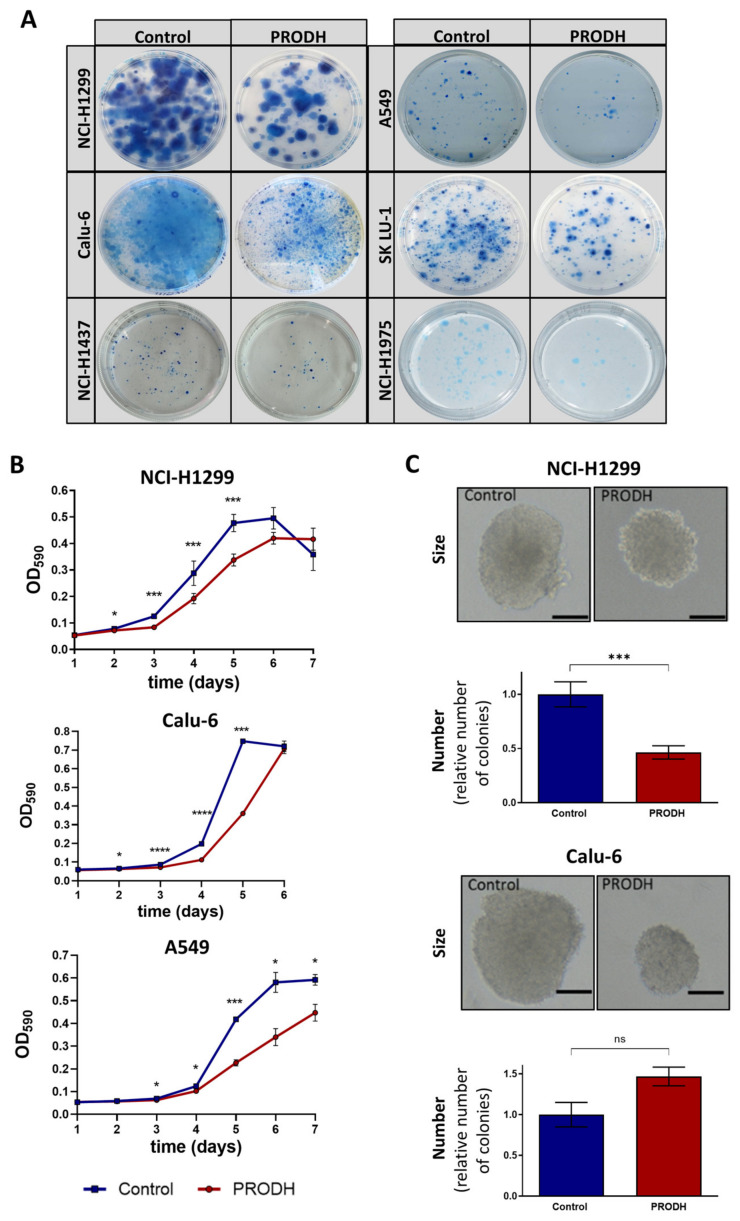
PRODH modulation affects survival and growth of lung ADC cell lines. (**A**) Clonogenic assays. The pictures show representative clonogenic assays carried out in the NCI-H1299, A549, Calu-6, SK LU-1, NCI-H1437 and NCI-H1975 cell lines. Petri dishes with 60 mm diameter were used for the experiments. Control: cells were transfected with pcDNA3.1 vector; PRODH: pcDNA3.1-PRODH (PRODH overexpression). (**B**) Growth curves. 3-(4,5-dimethylthiazol-2-yl)-2,5-diphenyltetrazolium bromide (MTT) assay was used to evaluate cell growth of PRODH-expressing or control clones from the indicated cell lines. The curves are the average of four PRODH-expressing clones and two control clones from the NCI-H1299 cell line, one PRODH-expressing and one control clone from the A549 and Calu-6 cell lines. Asterisks indicate significant differences in the growth of the two types of clones (Student’s *t*-test; * indicates *p*-value < 0.05; *** indicates *p*-value < 0.001; **** indicates *p*-value < 0.0001). (**C**) Soft agar assay. Representative pictures of colonies obtained with control or PRODH-expressing clones. The bar indicates 150 µm. For each clone tested, cells were plated at very low seeding density (5 × 10^2^ cells/mL) and grown in soft agar for 16 days. The colonies were counted for each clone and each replicate and the average number of colonies obtained from PRODH-expressing clones was plotted relative to the number of colonies obtained for control clones. The data represent the mean of triplicates ± standard error of the mean (SEM) of two independent experiments, using seven PRODH-expressing clones and seven control clones for NCI-H1299, or two PRODH-expressing and two control pools for Calu-6 cells. Results were analyzed by two-tailed Student’s *t*-test. Asterisks indicate significant differences (*** *p*-value < 0.001); “ns” indicates not significant difference.

**Figure 4 ijms-25-00714-f004:**
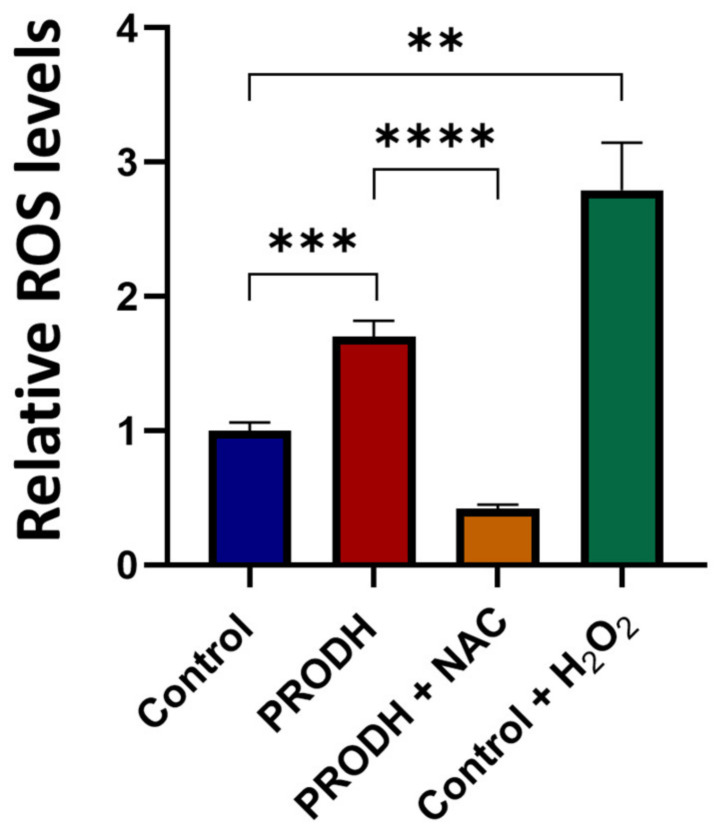
PRODH expression increases reactive oxygen species (ROS) production in NCI-H1299 cells. ROS measurement by 2′,7′-dichlorofluorescein diacetate (DCFDA) assay in PRODH-expressing and control clones from the NCI-H1299 lung ADC cell line. Four PRODH-expressing clones and two control clones were used. The average ± SEM of ROS level of each type of clone (PRODH-expressing and control clones) was calculated. ROS production by PRODH-expressing clones is shown relative to control clones. N-acetylcysteine (NAC) treatment neutralized the ROS increase observed in PRODH-expressing clones compared to control clones. H_2_O_2_ treatment was used as a positive control. Results were obtained from three independent experiments. Asterisks indicate significant differences in the relative ROS levels under the different conditions (Student’s *t*-test; ** indicates *p*-value < 0.01; *** indicates *p*-value < 0.001; **** indicates *p*-value < 0.0001).

**Figure 5 ijms-25-00714-f005:**
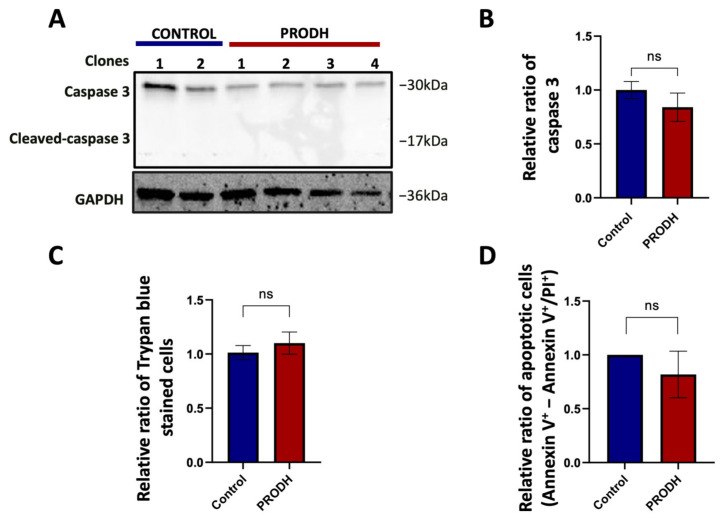
PRODH does not induce apoptosis in the NCI-H1299 cell line. (**A**) Analysis of caspase 3 and cleaved caspase 3 in PRODH-expressing and control clones. Analysis of caspase 3 and cleaved caspase 3 was carried out by immunoblotting. Levels were normalized to glyceraldehyde-3-phosphate dehydrogenase (GAPDH), using ImageJ software. (**B**) Quantification of Western blot data. Graph showing the average of caspase 3 protein normalized levels ± SEM in four PRODH-expressing clones relative to two control clones. The difference between control and PRODH-expressing clones was not significant (Student’s *t*-test). (**C**) Vital count with Trypan blue. The relative ratio of the average of Trypan blue stained cells ± SEM of four PRODH-expressing clones compared to the average of two control clones is shown. The difference between control and PRODH clones was not significant (chi-squared test). (**D**) Flow cytometric analysis of Annexin V/Propidium iodide (PI)-stained cells. The relative ratio of apoptotic cells (Annexin V^+^-Annexin V^+^/PI^+^) ± SEM of three PRODH-expressing clones relative to a control clone is shown. The difference between control and PRODH-expressing clones was not significant (Student’s *t*-test); “ns” indicates not significant difference. The uncropped original western blot has been submitted in the Appendix A.

**Figure 6 ijms-25-00714-f006:**
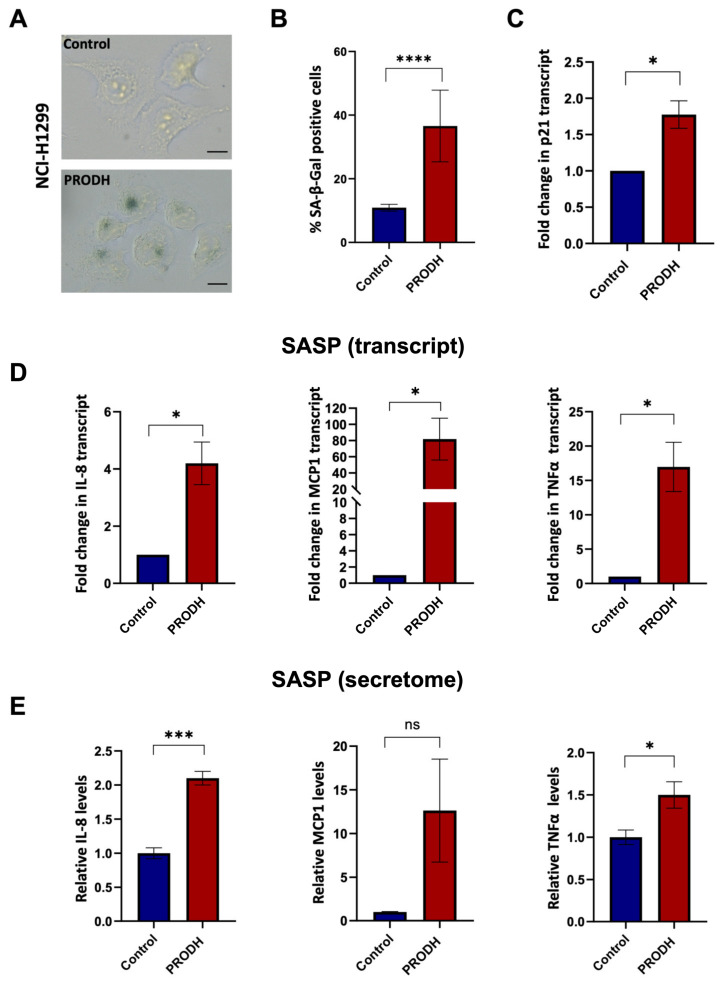
PRODH expression has a role in the induction of cellular senescence and senescence-associated secretory phenotype (SASP) in the NCI-H1299 lung ADC cell line. (**A**) Representative images of senescence-associated-β-galactosidase (SA-β-gal) staining in a control (up caption) and a PRODH-expressing clone (down caption). Magnification 400×. The bar indicates 14 μm. (**B**) Bar graph showing the percentage of SA-β-gal-positive cells. Average of the percentage of senescent cells ± SEM in the two control clones and the four PRODH-expressing clones is shown. Asterisks indicate significant differences in the two types of clones (chi-squared test; **** indicates *p*-value < 0.0001). (**C**) Analysis of p21 (cell cycle inhibitor) expression by qPCR. The graph shows the average ± SEM of qPCR results obtained in three experiments for three PRODH-expressing clones; the values are expressed relative to the average results of two control clones. Asterisks indicate significant differences between the two types of clones (Student’s *t*-test; * indicates *p*-value < 0.05). (**D**) Transcript levels of SASP genes. Transcript levels of interleukin-8 (IL-8), monocyte chemoattractant protein-1 (MCP-1) and tumor necrosis factor alpha (TNFα), in control and PRODH-expressing clones in NCI-H1299 cell line as detected by qPCR. The graphs show the average ± SEM of qPCR results obtained for three PRODH-expressing clones in three experiments for IL-8 and TNFα and in two experiments for MCP1, and the reported values are relative to the average results of two control clones. Asterisks indicate significant differences in the expression levels (Student’s *t*-test; * indicates *p*-value < 0.05). (**E**) Secreted levels of SASP soluble factors. Secreted protein levels of IL-8, MCP-1, and TNFα in control and PRODH-expressing clones of the NCI-H1299 cell line, as detected by secretome analysis. The graphs show the average ± SEM of secretome results obtained from two PRODH-expressing clones and two control clones, in NCI-H1299 cells. Asterisks indicate significant differences between the two types of clones (Student’s *t*-test; * indicates *p*-value < 0.05; *** indicates *p*-value < 0.001); “ns” indicates not significant difference. The uncropped original blots relative to secretome analysis and their elaboration have been submitted in the Appendix A.

**Table 1 ijms-25-00714-t001:** Correlation between PRODH expression and clinico-pathological data in NSCLC.

		NSCLC ^a^	
		ADC (70 Cases)	SCC (65 Cases)
		PRODH + Cases/Total (%)	PRODH + Cases/Total (%)
N. of cases		40/70 (57.14)	9/65 (13.85)
Grade	1	5/6 (83.33)	0/2 (0)
	2	24/50 (48)	8/40 (20)
	3	10/14 (71.43)	1/23 (4.35)
pT	1	22/32 (66.67)	5/26 (19.23)
	2	18/31 (58.06)	3/31 (9.68)
	3	0/5 (0)	1/4 (25)
	4	0/1 (0)	0/2 (0)
	x	0	0/2 (0)
pN	0	35/53 (66.04)	8/54 (14.81)
	1	0/2 (0)	1/2 (50)
	2	3/8 (37)	0/3 (0)
	x	2/7 (28.57)	0/6 (0)
Stage	I	37/55 (67.27)	8/55 (14.55)
	II	0/1 (0)	0/1 (0)
	III	3/8 (37.5)	1/6 (16.67)
	IV	0/5 (0)	0/3 (0)
	x	0/1 (0)	0

^a^ NSCLC: Non-Small Cell Lung Cancer.

**Table 2 ijms-25-00714-t002:** Cell lines used in this work.

Cell Line	Histology	Growth Medium	Origin
A549	NSCLC, adenocarcinoma	RPMI1640 ^a^ + 10% FBS ^b^ + 2 mM L-Gln	ATCC
NCI-H1299	NSCLC, lymph node metastasis	RPMI1640 + 10% FBS + 2 mM L-Gln	ATCC
NCI-H1975	NSCLC, adenocarcinoma	RPMI1640 + 10% FBS + 2 mM L-Gln + Sodium Pyruvate	Dr. A. Bisio, Università di Trento, Trento, Italy
Calu-6	NSCLC, anaplastic carcinoma (derived from metastatic site: pleural effusion)	RPMI1640 + 10% FBS + 2 mM L-Gln	Dr. E. Grassilli, Università di Milano Bicocca, Milan, Italy
NCI-H2228	NSCLC, adenocarcinoma	RPMI1640 + 10% FBS + 2 mM L-Gln + Sodium Pyruvate	Dr. E. Grassilli, Università di Milano Bicocca, Milan, Italy
SK LU-1	NSCLC, adenocarcinoma	DMEM ^c^ + 10% FBS + 2 mM L-Gln + Sodium Pyruvate + MEM ^d^ NEAA ^e^	Dr. E. Grassilli, Università di Milano Bicocca, Milan, Italy
NCI-H441	NSCLC, papillary adenocarcinoma	RPMI1640 + 10% FBS + 2 mM L-Gln + 10 mM Hepes + Sodium Pyruvate + Glucose (final 4500 mg/L)	Dr. V. Dall’Asta, Università di Parma, Parma, Italy
LX1	NSCLC, squamous-cell carcinoma	DMEM + 10% FBS +2 mM L-Gln	Cell Bank Interlab Cell Line Collection (ICLC), IRCCS San Martino Policlinico Hospital
SKMES-1	NSCLC, squamous-cell carcinoma (derived from metastatic site: pleural effusion)	DMEM + 10% FBS + 2 mM L-Gln	Cell Bank Interlab Cell Line Collection (ICLC), IRCCS San Martino Policlinico Hospital
HCC827	NSCLC, adenocarcinoma	RPMI 1640 + FBS 10% + 2 mM L-Gln	Dr. R. Alfieri, Università di Parma, Parma, Italy
HCC827-GR5	NSCLC, adenocarcinoma (derived from HCC827)	RPMI 1640 + FBS 10% + 2 mM L-Gln + 1 μM Gefitinib	Dr. R. Alfieri, Università di Parma, Parma, Italy
NCI-H2342	NSCLC, adenocarcinoma	DMEM:F12 Medium + heat inactivated FBS 5% + 4.5 mM L-Gln + 0.0005 mg/mL Insulin + 0.01 mg/mL Transferrin + 30 nM Sodium selenite + 10 nM Hydrocortisone + 10 nM beta-estradiol	Dr. G. Damia, Mario Negri Institute, Milan, Italy
NCI-H1437	NSCLC, adenocarcinoma (derived from metastatic site: pleural effusion)	RPMI 1640 + FBS 10% + 2 mM L-Gln	Dr. G. Damia, Mario Negri Institute, Milan, Italy
NCI-H727	Bronchial carcinoid	RPMI 1640 + FBS 10% + 2 mM L-Gln	Cell Bank Interlab Cell Line Collection (ICLC), IRCCS San Martino Policlinico Hospital
IGROV-1	Ovarian endometrioid adenocarcinoma (control for PRODH expression)	RPMI 1640 + FBS 10% + 2 mM L-Gln + MEM NEAA	ATCC

^a^ RPMI: Roswell Park Memorial Institute; ^b^ FBS: Fetal bovine serum; ^c^ DMEM: Dulbecco’s modified Eagle’s medium; ^d^ MEM: minimal essential medium; ^e^ NEAA: non-essential amino acids.

## Data Availability

Data is contained within the article and Appendix A.

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
