# Peer review of "Proline Dehydrogenase (PRODH) Is Expressed in Lung Adenocarcinoma and Modulates Cell Survival and 3D Growth by Inducing Cellular Senescence"

_ijms, 2024, doi:10.3390/ijms25020714_

Round 1
Reviewer 1 Report
Comments and Suggestions for Authors
1. Can you use TCGA database to investigate whether PRODH expression is associated with survival?
2. May be you can do some in vivo mouse tumor-bearing experiments to investigate PRODH-expression tumor growth in vivo.
3. Adding more discuss the underline mechanism of PRODH-mediated tumor inhibition is much more informative.
Comments on the Quality of English LanguageMinor reversion
Author Response
Dear Reviewer,
you will find the point-to-point reply in the attached file.
Thank you for revising our manuscript.

Reviewer 2 Report
Comments and Suggestions for Authors
In this work, the researchers seek to answer whether the enzyme proline dehydrogenase has a role in non-small cell lung tumorigenesis, as has been seen in other types of tumors, and to elucidate these functions. Likewise, they seek to demonstrate that this protein could function as a prognostic marker for early lung cancer.
The article addresses a controversial topic in the field of cancer; as the same authors mention in their background, proline dehydrogenase plays dual roles in different types of cancer; on the one hand, it can increase carcinogenesis, and on the other, it can cause an increase in apoptosis and autophagy which decreases cell survival. Therefore, this manuscript would be added to the list of articles that seek to clarify the role of proline dehydrogenase in cancer, particularly lung cancer, which declares it a relevant article in its field.
There is little data on the role of proline dehydrogenase in lung cancer; in fact, the little published (including the article by Liu et al. 2020) supports the idea that proline dehydrogenase increases carcinogenesis, including metastasis. However, the current article opens the debate on the dual role of this enzyme, which has already been established in different articles. The results of this article are more aimed at supporting the “protective” role of proline dehydrogenase expression since a decrease in cell survival is observed, possibly by increasing cell senescence.
Liu, Y., Mao, C., Wang, M. et al. Cancer progression is mediated by proline catabolism in non-small cell lung cancer. Oncogene 39, 2358–2376 (2020).
The methodology of this study is well written and complete; However, to fulfill the objective of demonstrating whether proline dehydrogenase is a good marker for early lung cancer, non-cancerous lung cells should be included as a control to establish the difference between the proline dehydrogenase levels of a healthy cell and a cancer cell. With these data, we could better predict whether or not proline dehydrogenase could be an excellent prognostic marker. On the other hand, the loading controls for the Western blot analysis must be improved since a significant difference is seen in the bars, particularly those used in Figure 2.
The general conclusion of the article addresses the initial question and is supported by the results. In contrast, the conclusion in the abstract promises a better discussion of proline dehydrogenase as a prognostic marker. However, the article is poor in this information. There is not much discussion, and the only results demonstrate that this enzyme is highly expressed in early tumors, so a comparison of the levels with non-cancerous cells and a better discussion is needed.
The references are appropriate, no misused self-citations were detected, and relevant articles in the field are cited.
Abbreviations need to be defined in the summary.
Line 67: Improve the wording of the first statement. “The catabolism of proline occurs exclusively through proline dehydrogenase (PRODH, EC 1.5.5.2, formerly EC 1.5.99.8), a mitochondrial inner-membrane and stress-inducible enzyme containing a flavin adenine dinucleotide cofactor (FAD)”.
Figure 1: Change the panels of Figure 1A to numbers to distinguish them from the panels of the figure.
Figure 2: Improve loading control for Western blots, particularly for NCI-H727 cells.
Abbreviations should be used correctly. They should be defined the first time used and repetitions should be avoided. In additions abbreviations used in tables and figures should be defined in the respective legend or footnote. Figures and tables should be understable without reading the main text.
Examples, these abbreviation should be defines the firts used:
ADCs line 30, EGFR line 33, NCI-H1299 line 34, NCI-H1299 line 36, Calu-6 line 36
Examples, Repetitions should be avoided
PRODH line 67, PRODH line 331, SCLC line 436, SASP line 695, DELTA Ct line 713
Corrections:
1. Table 2: All the acronyms used (e.g. RPMI, DMEM, etc.) in the table should be defined at the bottom (as a foot note)
2. Line 575: Delete space between “...2 x …”
3. Figure 2: Should be defined “PRODH, ddPCR, RNA, cDNA, PRODH”
4. Line 152: Delete space between “asterisk and p=...”
5. Line 162: Delete space between “p= and 0.0099”
6. Figure 3: Should be define “PRODH, MTT”
7. Line 238: Delete space between “(asterisk and p=...” and “...p < 0,001)”
8. Figure 4: Should be define “...ROS, DCFDA, PRODH, NAC, H2O2…”
9. Line 257: Delete space between “...p-value < 0,00…)” in all 3 cases.
10. Figure 5: Should be define PRODH, GAPDH, NCI-H1299
11. Line 293: Delete space between “...p < 0.0001…”
12. Figure 6: Define PRODH, SASP, NCI-H1299, SA-β-gal, qPCR, IL-8, MCP-1 and TNFα
13. Line 309: Delete space between “...p-value < 0,0001…”
14. Line 412: Delete space between “...(age >65 years)...”
15. Line 466: Delete space between “...TBS + 0,2%...”
Author Response

(The authors gave the same response as above.)

Reviewer 3 Report
Comments and Suggestions for Authors
This is a good and complex study linking proline dehydrogenase expression in lung adenocarcinoma. The authors report that the enzyme takes place in cellular senescence.
Add information on used techniques to the Abstract
Lines 101-102 are a description of the results, not the extended aim of the study
Provide concentrations of use antibodies
Describe changes in the morphology of cells in Fig. 6A
Provide the passage number of each cell line
Add study limitations
Author Response

(The authors gave the same response as above.)

Round 2
Reviewer 2 Report
Comments and Suggestions for Authors
The authors addressed carefully the observations of this reviewer